# Effect of integrated reporting quality disclosure on cost of equity capital in developed markets: Exploring the moderating role of corporate governance quality

**Muhammad Arslan Iqbal**[1], **Md Abdur Rouf Sarkar**[2,3*,], **Majed Alharthi**[4], **Md Jahid Ebn Jalal**[2], **Md. Naimur Rahman**[5,6,7]

1 School of Accounting, Zhongnan University of Economics and Law, Wuhan, China, 2 School of Economics, Zhongnan University of Economics and Law, Wuhan, China, 3 Agricultural Economics Division, Bangladesh Rice Research Institute, Gazipur, Bangladesh, 4 Finance Department, College of Business, King Abdulaziz University, Rabigh, Saudi Arabia, 5 Department of Geography, Hong Kong Baptist University, Kowloon, Hong Kong, 6 David C Lam Institute for East-West Studies, Hong Kong Baptist University, Kowloon, Hong Kong, 7 Department of Development Studies, Daffodil International University, Dhaka, Bangladesh.

* mdrouf_bau@yahoo.com

## Abstract

This study examines the relationship between integrated reporting quality (IRQ) disclosures, corporate governance quality (CGQ), and the implied cost of equity capital (ICC) in developed markets, focusing on Australia and New Zealand. The increasing adoption of integrated reporting and its potential implications for firms' ICC motivates this research. Moreover, the study highlights the role of IRQ in mitigating information asymmetry between firms and investors, emphasizing the need for high-quality disclosures. Using a quantitative approach with panel data analysis, the research analyzes a sample of the top 174 companies by Standard and Poor's market capitalization in Australia and New Zealand from 2018 to 2022, encompassing 870 observations post-IRQ implementation. Statistical methods, including fixed-effects, IV2SLS, two-step system-GMM, pooled OLS, and medium quantile regression, were applied to ensure robust findings. The results reveal a significant negative relationship between IRQ disclosure and ICC, with CGQ playing a moderating role in strengthening this association. Consistent with agency theory, the findings suggest that to reduce information asymmetry, firms issue more information which allows to reduce the cost of capital. Therefore, a more comprehensive firms' reporting, including information about their strategy and risks, increases investors' confidence, hence it may reduce the cost of capital. This study provides valuable insights for regulators and policymakers by emphasizing the importance of integrated reporting frameworks and robust corporate governance practices to promote transparency, reduce information asymmetry, and optimize capital allocation efficiency in developed markets.

**Data availability statement:** All relevant data are within the manuscript and its Supporting Information files

**Funding:** The author(s) received no specific funding for this work.

**Competing interests:** The authors have declared that no competing interests exist.

## 1. Introduction

The proliferation of integrated reporting (IR) has sparked considerable scholarly interest in recent years [1–4]. Stakeholders, particularly shareholders, now regularly demand comprehensive financial information. Nonetheless, deficient financial reporting can detrimentally impact the interpretation of business performance [5,6]. Key criteria embraced by investors and stakeholders include the utilization of IR within the capital market to furnish comprehensive disclosure in a single report [7]. As previously indicated, IR facilitates stakeholders in grasping the essence of a company, its operations, and its performance [1,8]. It further affords investors and other stakeholders' insight into business management practices, thereby enabling them to anticipate inherent risks [9]. Integrated reporting represents a novel corporate reporting approach that consolidates diverse forms of information, whether financial or non-financial, into a unified report. Its aim is to enrich annual reports by furnishing more detailed financial insights into present and future prospects [1,10–12]. According to the framework established by the International Integrated Reporting Council (IIRC), the primary objective of IR is to "enhance the quality of information available to financial capital providers, thereby facilitating more efficient and effective capital allocation" [13]. The practice of IR strives to deliver superior quality information rather than sheer volume, aligning with investors' growing preference for enhanced decision-making resources [14]. An IR seeks to present a lucid and succinct portrayal of an organization, its strategies, and risks, interconnecting its financial and sustainability performance, and providing stakeholders with a holistic and exhaustive comprehension of the organization and its future outlook [15,16].

Conventional corporate reporting fails to adequately satisfy stakeholders' information requirements for evaluating both historical and prospective company performance [4,17,18]. In recent times, companies have surpassed fundamental legal obligations for corporate social responsibility (CSR) reporting, as highlighted by Schmeltz [19] and Waddock [20]. Sustainability reporting emphasizes the dissemination of data concerning a company's performance across environmental, social, and governance dimensions, whereas IR underscores the strategic amalgamation of this data with other pivotal performance indicators to offer a coherent depiction of how the company generates value for investors [16,21]. Organizations adopting IR compile reports that amalgamate financial, environmental, social, and governance information in a cohesive manner [22,23]. Information gaps between companies and investors frequently arise due to management's selective disclosure practices, leading to conflicts and distrust, thereby causing investors to exhibit reluctance in investing in specific companies. Consequently, investors may demand an equity premium to allocate capital to such companies [24], thereby elevating the cost of equity capital [25–28], which bears significance in companies' capital budgeting and other financial planning activities [29]. The interplay among firm value, disclosure practices, and cost of equity capital is salient, with the cost of capital exerting a pivotal influence on a company's valuation. A higher cost of equity capital implies lower long-term returns for a company, potentially diminishing investors' perceived value in holding a stake in said

company. Notably, the degree of integration of environmental, social, and governance (ESG) disclosures can markedly impact firm value. Sun et al. [30] identified that the negative correlation between the integration level of ESG disclosures and firm value stems from challenges associated with the comprehensibility of these disclosures.

The relationship between integrated reporting quality (IRQ) disclosure and the implied cost of equity capital (ICC) has been widely examined; however, prior research has yielded mixed and inconclusive findings [31–36]. While there is evidence of a correlation between an integrated reporting framework and capital market outcomes [35–37], the voluntary nature of IR [38] has not hindered its widespread adoption, particularly among large corporations in developed nations [11]. Despite the growing prominence of IR, limited attention has been devoted to the moderating role of corporate governance quality (CGQ), even though CGQ is a critical factor influencing investor confidence and firm performance [9,37,39,40]. Our study addresses this gap by investigating the interplay between IRQ, CGQ, and ICC within developed markets, focusing on Australia and New Zealand. These markets are distinguished by their stringent regulatory frameworks, robust investor protection mechanisms, and efficient capital markets [41], making them an ideal setting for exploring how governance quality and reporting practices jointly influence financial outcomes.

This research contributes to the literature in three key ways. First, it integrates CGQ as a moderating variable, offering new insights into how governance frameworks amplify the benefits of integrated reporting on ICC. Second, it employs advanced econometric methods, such as system-GMM and IV2SLS, to address endogeneity concerns, ensuring robust and reliable findings. Third, it extends agency theory by demonstrating how IR reduces information asymmetry, particularly in firms with weaker governance structures, thereby enhancing capital efficiency.

The principal aim of this study was to investigate the moderating role of CGQ in augmenting the relationship between IRQ disclosure and ICC in developed markets. By focusing on top-listed firms in Australia and New Zealand, this study provides actionable insights for companies, investors, and policymakers aiming to optimize reporting and governance practices.

The article is structured as follows: Section 2 provides an overview of the research context, including a review of relevant literature. Section 3 describes the study area, data, and empirical methodologies. Section 4 presents the findings and discussion, while Section 5 summarizes the conclusions and proposes directions for future research.

## 2. Research in context

### 2.1 Relationship between IRQ disclosure and ICC

Integrated reporting has gained attention as a mechanism for improving financial transparency and aligning corporate governance with stakeholder expectations [1,11,12,36,42]. By integrating financial and non-financial disclosures, IR aims to enhance decision-making, reduce information asymmetry, and ultimately lower the cost of capital [36]. Non-financial data, including CSR initiatives, have been observed to correlate with the cost of equity capital. Breuer et al. [43] investigated the link between CSR, investor protection, and equity capital costs, noting that in jurisdictions with robust investor safeguards, CSR investment tends to lower equity costs, whereas in contexts with weaker investor protection, such investments tend to elevate equity costs.

Theoretical frameworks, as elucidated by Dhaliwal et al. [44], underscore the significance of employing prospective information to gauge equity capital costs. Voluntary disclosure of non-financial data is viewed as a means to bridge the gap between external and internal information, thereby diminishing equity capital costs and attracting long-term investors. Moreover, following the mandatory adoption of international financial reporting standards (IFRS), there was an enhancement in accounting information quality [1,45–47], leading to a subsequent reduction in capital costs [9,48,49]. Enhanced information quality is associated with fewer instances of managerial misappropriation, thereby resulting in lower equity capital costs [37,50]. Nonetheless, certain studies have identified a negative correlation between voluntary non-financial information disclosure and equity capital costs [35,51,52].

Integrated reporting represents a voluntary reporting framework that holds promise for revolutionizing corporate reporting practices [36,53]. Recent data from a survey conducted by PricewaterhouseCoopers (PwC) reveals that approximately two-thirds of investment professionals perceive the quality of a company's reporting, encompassing details about strategy, risks, and other value drivers, as directly influencing the cost of capital [1]. Nonetheless, a dearth of studies exists investigating the nexus between IRQ and the capital market, warranting further exploration in this domain to enhance comprehension of integrated reporting's impacts on the cost of equity capital.

Prior studies have explored the relationship between the quality of financial disclosures and a firm's cost of capital [35,54,55]. Research using global datasets analyzed IR and capital costs across 27 countries, covering 995 firms and 3,294 observations from 2009 to 2013 [33,34]. Maria and Ligia's [33] and Chouaibi et al. [35] found a negative association between IRQ disclosure and the cost of capital but noted that differences in corporate governance systems among countries did not significantly influence this relationship. In contrast, Barth et al. [32], studying 80 firms listed on the Johannesburg Stock Exchange, found no significant relationship between IRQ and ICC. Similarly, Barth et al. [32] and Lee and Yeo [31] observed a positive association between IRQ disclosure and firm valuation, contributing to mixed findings on the IRQ-ICC relationship.

Existing literature also highlights limitations. For instance, while de Villiers et al. [11] primarily focused on the conceptual underpinnings of IR, emphasizing its potential to provide a comprehensive narrative of value creation but lacking empirical insights into its impact on cost of capital. Pavlopoulos et al. [42] identified a positive association between IR quality and corporate governance but did not explore how this interaction affects financial outcomes like cost of equity. Similarly, Gupta et al. [54] demonstrated the significance of governance mechanisms and financial development in reducing cost of equity but did not examine IR as a specific governance tool. This leaves a critical gap in understanding how IRQ interacts with CGQ to influence a firm's financial outcomes, particularly within different institutional contexts.

Our study aims to address this gap by investigating the direct and interactive effects of IRQ and CGQ on the cost of capital. Unlike prior research, our approach explicitly incorporates the dynamic interplay between these variables, offering insights into their complementary roles across diverse regulatory and financial environments. By doing so, this study contributes to a more nuanced understanding of the mechanisms through which IR impacts financial metrics, expanding the scope of existing literature.

## 2.2 Relationship among IRQ disclosure, CGQ, and ICC

Previous investigations have provided empirical support for the advantages associated with the adoption of high-quality integrated reporting [1,9,13,56,57]. However, an exclusive focus on IRQ may not furnish a comprehensive understanding of its influence on capital market outcomes, particularly its efficacy on the cost of capital, without accounting for a robust institutional framework. Accounting practices, as evidenced in prior studies, were shaped by diverse factors, including the underlying capital markets [58–61]. While certain studies have suggested a tenuous relationship between IRQ disclosure and ICC [31–34], there exists a correlation between an integrated reporting framework and the capital market [37,62].

Nonetheless, extant research has predominantly concentrated on the impact of high-quality disclosure on the capital market, overlooking other institutional variables, despite theoretical assertions regarding the synergy between high-quality reporting disclosure and other institutional factors. Previous study has illustrated that CGQ has been employed as a moderating variable in numerous investigations [63–68].

Our research endeavors to bridge this void by exploring the influence of corporate governance on both the IRQ disclosure and ICC. Through the examination of employees' role in CGQ as a moderating variable, we aim to offer valuable insights into how governance practices shape the relationship between IRQ disclosure and ICC. In this segment, we delve into the correlation between IRQ disclosure and ICC (Section 2.1). Subsequently, we broaden our investigation to encompass the association between IRQ disclosure, CGQ, and ICC (Section 2.2). Previous studies have already scrutinized the nexus between voluntary disclosure, encompassing both financial and non-financial aspects, and corporate governance [69–74].

A pioneering approach to enhancing CSR reporting is IR, which endeavors to amalgamate CSR disclosures [75]. Furthermore, scholars have noted a positive linkage between CGQ and CSR, with Huang [76] documenting a robust correlation. Similarly, in emerging economies, Zaid et al. [77] have affirmed the influence of CGQ practices on CSR disclosures. However, Sriani and Agustia [78] identified no significant association between IRQ and information asymmetry as measured by the spread. The ESG information is pivotal in evaluating companies' risk profiles, future cash flows, and effectively pricing investments [79]. Institutional factors, such as corporate governance, directly shape voluntary disclosure and its ramifications on equity costs [12,42,54,80,81].

Hence, drawing upon extant literature, we envisage that CGQ assumes a pivotal role in fortifying the relationship between IRQ disclosure and ICC in developed markets. De Villiers et al. [11] introduced IRQ as an approach to reporting that encompasses both financial and non-financial information within a unified report. Subsequent extensive research has investigated the correlation between IRQ disclosures, CGQ, and ICC. Consistently, studies have revealed a positive association between IRQ disclosure and these variables [70]. Pavlopoulos et al. [42] contend that high-quality disclosure via integrated reporting fosters more effective corporate governance mechanisms. Consequently, this influences companies' investment decisions, impacting the anticipated cash flow ratio and ultimately influencing the cost of equity capital [82].

Indeed, effective CGQ plays a pivotal role in mitigating both agency costs and ICC [80,83]. Corporate governance mechanisms are crafted to address agency dilemmas and ensure managerial actions align with shareholders' best interests [83,84]. The theoretical underpinning of the nexus between corporate governance and disclosure stems from agency theory, which tackles issues of information asymmetry and accountability, thereby resulting in a reduction in the cost of equity capital. Through the implementation of robust corporate governance practices, firms can bolster transparency and accountability, consequently diminishing information asymmetry between managers and shareholders. This engenders increased confidence among investors and analysts, positively shaping perceptions regarding the firm's value and future prospects. Consequently, the ICC diminishes as investors perceive the firm as less risky, thereby demanding a reduced return on their investment.

When integrated with robust CGQ, IRQ disclosure can reinforce these favorable outcomes. IRQ disclosure, encompassing both financial and non-financial information, offers a comprehensive overview of a firm's performance and sustainability endeavors. This can lead to enhanced decision-making, heightened stakeholder confidence, and improved long-term strategizing, all contributing to the reduction of the ICC. Overall, the research underscores the notion that a combination of robust CGQ and IRQ disclosure can exert a positive influence on the ICC. Through augmenting information quality, aligning managerial and shareholder interests, and fostering transparency, firms can cultivate a conducive investment environment that attracts investors and ultimately diminishes the ICC.

The literature consistently highlights the interconnection between CGQ and ICC, with various conceptualizations of corporate governance demonstrating an influence on the ICC. Empirical investigations consistently indicate that firms implementing stronger governance practices experience diminished capital costs [85,86]. Furthermore, Gupta et al. [54] evidenced that firm-level CGQ amplifies the effect on the ICC. Conversely, Boubakri et al. [87] observed that distant privatized firms in countries characterized by weak institutional governance encounter elevated ICC. Zhou [37] unveils that firms boasting robust CGQ incur lower costs of equity, particularly in nations with robust legal frameworks, sound government quality, and extensive disclosure standards. Jebran and Chen [88] analyze how corporate governance practices aided firms in navigating the challenges posed by the COVID-19 pandemic. Specifically, Pavlopoulos et al. [42] establish a positive correlation between IR and CGQ. Nonetheless, it is noteworthy that the literature presents mixed findings regarding the relationship between CGQ and ICC [54,80,89].

Given this diversity in results, CGQ plays a pivotal role in influencing a firm's cost of capital by enhancing transparency, mitigating agency conflicts, and improving investor confidence [54]. Existing literature highlights that strong governance practices, such as board independence and robust shareholder rights, contribute to lower capital costs, particularly in regions with developed financial systems. Pavlopoulos et al. [42] extended this discourse by demonstrating that firms with

higher IRQ exhibit superior governance mechanisms, which further reduce agency costs. However, they did not explore how these governance improvements mediate the relationship between IRQ and cost of capital. Gupta et al. [54] emphasized the complementary effects of country-level financial development and firm-level governance in reducing cost of equity, underscoring the importance of external institutional environments. Building on this, we argue that IRQ can act as a catalyst for enhancing CGQ, thereby amplifying its impact on financial outcomes. While prior studies have largely treated CGQ and IR as independent constructs, this study examines their interaction, hypothesizing that IRQ enhances CGQ's effectiveness in reducing cost of capital by fostering integrated decision-making and improving the quality of disclosures. By investigating these interdependencies, this study not only fills a significant gap in the literature but also provides actionable insights for policymakers and practitioners seeking to leverage IR as a tool for improving governance and financial performance. The findings are expected to inform strategies for aligning IR and CGQ to achieve optimal financial outcomes, particularly in emerging markets where governance and disclosure practices are still evolving.

## 2.3 Theoretical framework

Fig 1 illustrates the path diagram depicting the relationships among IRQ disclosure, CGQ, and ICC. Theoretical perspectives strongly advocate that the nexus between disclosure policy, reporting quality, and the cost of capital significantly shapes investors' risk assessment by mitigating asymmetric information [90,91]. This mitigation of information asymmetry among investors [92,93] not only diminishes estimation risk but also augments transparency, ultimately culminating in a reduced cost of equity capital [34,36,94]. The established connection between accounting information and ICC in theoretical studies emphasizes that the extent of disclosure is intricately linked to the cost of capital [90]. Signaling theory, originally conceived to address the information disparity in the labor market [95], has been employed to elucidate voluntary disclosure in corporate reporting [96]. In an endeavor to bridge this information gap, companies strategically furnish information to investors, showcasing their superior performance vis-à-vis other market competitors, with the aim of attracting investments and fostering goodwill in the market [90]. Subsequently, this exerts an influence on the cost of equity capital. Overall, these theoretical frameworks underscore the pivotal role played by disclosure practices and reporting quality in mitigating information asymmetry, shaping investors' risk perceptions, and ultimately affecting the cost of equity capital for firms.

Previous research has delved into the correlation between IR and ICC, yielding inconclusive findings [31–36]. Integrated reporting entails the holistic integration of financial, non-financial, strategic, and governance data into a unified report. Considering the comprehensive nature of IR, it is anticipated that robust institutional factors, such as corporate governance, assume a pivotal role in fortifying the relationship between IRQ disclosure and ICC. These factors are

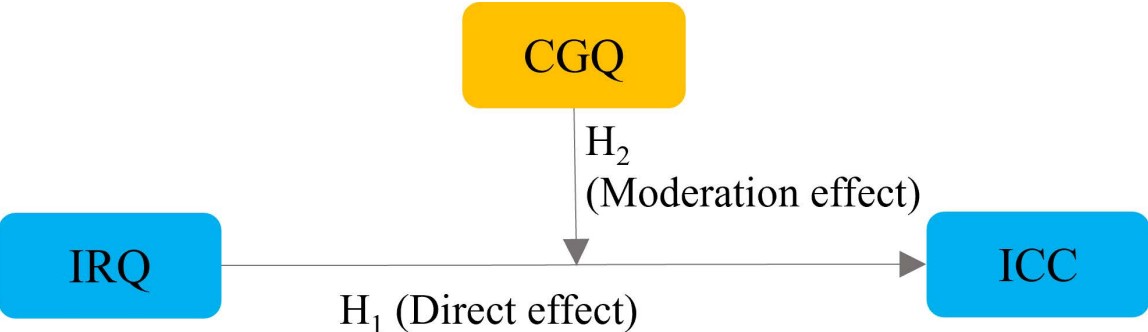

**Fig 1. Conceptual framework illustrating the moderating effect of CGQ on the relationship between IRQ disclosure and ICC. Here, CGQ, IRQ, and ICC denotes corporate governance quality, integrated reporting quality, and implied cost of equity capital, respectively.**

envisaged to bolster the positive impacts of disclosure quality on diminishing the cost of capital and subsequently enhancing investors' risk perception.

In the realm of accounting and finance research, agency theory serves as a prominent framework. Within agency theory, the owners are denoted as principals, while the managers act as agents [97]. Under the purview of agency theory, it is posited that firms will voluntarily disclose more information to mitigate agency costs, which arise from conflicts between stakeholders and managers [98]. Furthermore, agency theory underscores the pivotal role of corporate governance mechanisms, particularly the board of directors, in fostering robust CSR practices [4,99]. This concept was originally postulated by Jensen and Meckling [97] in their framework, establishing a direct linkage between information disclosure practices and the internal mechanisms of corporate governance. According to agency theory, firms implement effective control mechanisms to safeguard shareholders' interests and address conflicts of interest. Within this context, managers are expected to prioritize shareholders' interests over their personal interests [100,101], while the board of directors assumes the responsibility of overseeing the effective operation of corporate governance systems and ensuring the utmost protection of shareholders' interests [4,99,102].

Firms implementing IRQ components are associated with superior accounting disclosure quality, leading to decreased agency costs and provision of reliable reporting [32,56]. In our investigation, we assessed governance quality based on attributes such as anti-takeover provisions, audit quality, compensation and ownership structure, and board composition, anticipating a positive influence on IRQ disclosure and consequent reduction in the ICC. This observation aligns with Gupta et al. [54] findings, which underscore the pivotal role of corporate governance in bolstering financial development and lowering the ICC. Empirical evidence corroborates these assertions, as research by Skaife et al. [82] demonstrates a direct and indirect positive impact of enhanced corporate governance on reducing agency costs and diminishing the ICC. Furthermore, board independence and size have been linked to increased environmental and social disclosure [99,103], with independent directors serving as effective monitoring mechanisms for managers [82], thereby enhancing accounting disclosures and emphasizing the influence of board composition on corporate reporting [99,104–106]. Tran et al. [107] also noted positive effects of the audit committee, board size, and presence of foreign members on CSR disclosure. Additionally, Adnan et al. [81] and Devarapalli and Mohapatra [4] underscored the positive influence of CGQ, particularly through the presence of social responsibility board committees and IRQ disclosure, on augmenting CSR reporting. Similarly, Haldar and Raithatha [70] and Jarah et al., [108] identified significant enhancements in firms' financial disclosure practices with improved governance practices, particularly through the composition of the audit committee. Crucially, Pavlopolos et al. [42] uncovered a positive association between IRQ disclosure and corporate governance.

Considering the above, it is notable that the advantages stemming from decreased cost of equity capital as a result of enhanced governance practices may be more accessible to large enterprises in Australia and New Zealand with access to foreign capital markets. Conversely, smaller firms may not have equivalent opportunities to fully capitalize on these effects. The caliber of corporate governance is poised to play a pivotal role in augmenting the association between IRQ disclosure and ICC. Therefore, this study posits the following hypotheses:

**H₁.** There exists a negative correlation between IRQ disclosure and ICC.

**H₂.** The CGQ moderates the correlation between IRQ disclosure and ICC.

### 2.4 Control variables

To ensure unbiased findings, it is crucial to account for the influence of various factors on the ICC, including company size, long-term growth, return on assets (ROA), CSR, and leverage. These variables are often incorporated by researchers to elucidate the level of information disclosure, a key component of corporate transparency [4,16,33,35,99,109,110]. Company size (SIZE), measured as the natural logarithm of a firm's market value of equity [92], typically exhibits a negative coefficient in the relationship between information disclosure and ICC [50,111]. However, empirical studies on long-term growth (LTG) have produced mixed results, with some suggesting a positive association between ICC and LTG rate

[112–114], while others observe a negative sign [44]. Profitability is commonly considered in the context of sustainability initiatives and reporting, with an inverse relationship noted between profitability and information disclosure [4,35,115,116]. Nonetheless, findings regarding the significance of profitability in relation to the cost of capital are inconclusive [117]. Recent research on CSR reports indicates that issuing a stand-alone CSR report can enrich companies' information environment, leading to a reduction in ICC and improved analyst forecasts [44,118,119]. Therefore, the inclusion of a stand-alone CSR report issuance as a control variable in the model is anticipated, with an expected negative coefficient. Leverage (LEV), measured as the ratio of total debt to total assets, also plays a pivotal role in the voluntary disclosure of corporate information [57,120]. By controlling for these factors, we aim to isolate the specific moderating role of corporate governance between IRQ disclosure and ICC, thereby ensuring the accuracy and reliability of our study results (Fig 2).

## 3. Methodology

### 3.1 Study area and data

This study focuses on Australia and New Zealand, selected for their advanced regulatory frameworks, robust corporate governance standards, and early adoption of integrated reporting practices. These countries, characterized by mature markets and comprehensive corporate transparency initiatives, provide an ideal context for examining the interplay between IRQ, CGQ, and ICC. The sample selection involved purposively identifying the top 100 largest companies listed in both countries based on S&P market capitalization. The selected firms were required to practice IRQ disclosure and demonstrate robust governance frameworks.

Data for the study were collected from multiple sources, including publicly available databases, company annual reports, regulatory filings, and Institutional Shareholder Services (ISS) risk metrics. The dataset spans the period from 2018 to 2022, covering 870 observations from a sample of 174 firms over five years. To ensure data reliability, we cross-validated the collected information with secondary sources, such as annual integrated reporting data from the Australian and New Zealand stock exchanges, capital market data, official company websites, and Bloomberg. Financial variables—including ICC, leverage, firm market value of equity, long-term growth, return on assets, CSR, and the ratio

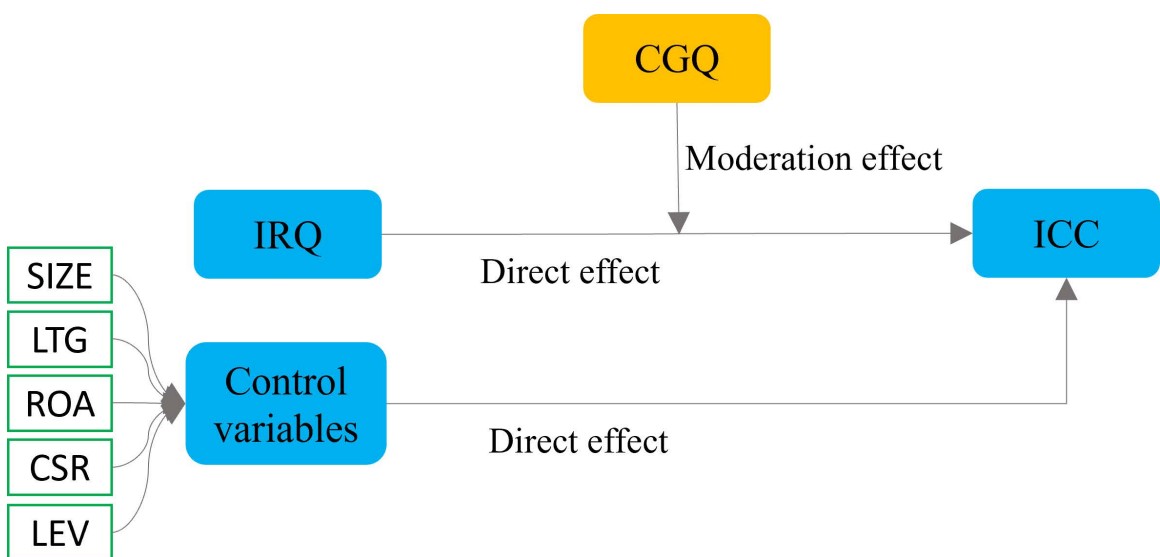

**Fig 2. Conceptual framework illustrating the moderating effect of CGQ on the relationship between IRQ disclosure and ICC (including control variables).** Here, CGQ, IRQ, ICC, SIZE, LTG, ROA, CSR, and LEV denotes corporate governance quality, integrated reporting quality, implied cost of equity capital, company size, long-term growth, return on assets, corporate social responsibility, and leverage, respectively.

of total debt to total assets—were sourced from Thomson Reuters Eikon and I/B/E/S within the DataStream databases. After data collection, we conducted several preprocessing steps to ensure robustness. These included detecting and removing outliers and addressing missing data using multiple imputation techniques, in line with established research practices [121].

### 3.2 Empirical settings

**3.2.1 Measurement of dependent variable.** In this research, the dependent variable was the implied cost of equity capital (ICC), a metric commonly employed in numerous prior studies [54,111,122–124]. ICC is computed as the average value using the rPEG model devised by Easton [125] for the twelve-month period subsequent to the end of the fiscal year. The estimation model is delineated as follows in Eq. (1):

$$ICC_{peg} = \sqrt{\frac{EPS_2 - EPS_1}{P_0}}$$

(1)

Where, $ICC_{peg}$ represents the implied cost of equity capital; $EPS_1$ and $EPS_2$ denote the one-year and two-year-ahead median analyst earnings forecasts per share, respectively; and $P_0$ signifies the daily price per share immediately preceding the EPS forecasts.

**3.2.2 Measurement of moderating variable.** In this research, we employed a methodology akin to that utilized by Aggarwal et al. [126,127] and Gupta et al. [54] to assess CGQ as an explanatory factor. Specifically, we relied on institutional shareholder services (ISS) risk metrics to appraise a firm's performance, drawing from regulatory filings, annual reports, and website content. ISS has devised a comprehensive rating framework encompassing four overarching categories: anti-takeover measures, audit practices, compensation and ownership structures, and board effectiveness [64,127]. Subsequently, we devised individual ratings for each of these categories, comprising board composition and efficacy, anti-takeover provisions, director and executive remuneration as well as ownership structures, and audit methodologies. These ratings were amalgamated to derive an overarching score for each firm, consistent with the methodology adopted by Aggarwal et al. [126,127].

Presenting CGQ as a singular variable, rather than delineating it across four categories, offers several advantages. Firstly, it streamlines the analysis and facilitates result interpretation by furnishing a unified score encapsulating a company's overall governance performance [54,127]. This simplification aids stakeholders, including investors and regulatory bodies, in juxtaposing and evaluating governance practices across diverse entities [37,85]. Secondly, amalgamating the four categories into one variable mitigates the risk of oversimplification, fostering a more holistic assessment of CGQ while reducing information asymmetry and enhancing stakeholder trust [127]. This approach acknowledges the interconnectedness of various governance facets, underscoring the holistic nature requisite for effective governance [42,80]. Finally, presenting CGQ as a singular variable fosters' transparency and accountability, furnishing stakeholders with a lucid and readily comprehensible metric of a company's governance practices. This transparency strengthens stakeholder confidence and enhances the firm's reputation, aligning with the broader objectives of good governance [69]. By consolidating the governance dimensions into a unified measure, our approach provides a practical and robust tool for assessing overall governance performance. This methodology aligns with prior research emphasizing the importance of integrated governance metrics in reducing complexity and improving comparability across firms [39,40,42,54,80,85,127].

**3.2.3 Measurement of IRQ disclosure variable.** Integrated reporting quality data were derived from content analyses of firms' annual reports and sustainability reports, following standardized disclosure frameworks such as the IIRC guidelines [128–131]. This checklist incorporated a structured scoring system ranging from '0' to '3'. Companies received a score of "0" for mere compliance, "1" for general qualitative disclosures, "2" for provision of specific information, and "3" for comprehensive discussions incorporating both qualitative and quantitative details. Its proven effectiveness in prior studies further strengthens its reliability and validity [16,99,132,133].

**3.2.4 Panel data model specification.** This research employs a panel data methodology to explore the association between the independent variables (namely, corporate governance quality and integrated reporting quality disclosure) and the dependent variable (i.e., cost of equity capital). A fixed or random effect model is utilized to address unobserved heterogeneity across individual observations and to mitigate potential endogeneity concerns. The Hausman test, a commonly employed technique in econometrics, is utilized to ascertain the suitability of employing either a fixed effect or random effect model in panel data analysis. Our study's Hausman test yielded a Chi-square value of 67.61 (p<0.01), indicating that the fixed effect model was more suitable than the random effect model (further elaborated in Table 3).

The fixed effect regression model utilized in the empirical analysis is as follows in Eq. (2):

$$ICC_{it} = \beta_0 + \beta_1 IRQ_{it} + \beta_2 CGQ_{it} + \beta_3 SIZE_{it} + \beta_4 LTG_{it} + \beta_5 ROA_{it} + \beta_6 CSR_{it}$$
$$+ \beta_7 LEV_{it} + \sum_{i=1}^{n} \beta_8 Company's\_Dummy_{it} + \sum_{i=1}^{n} \beta_9 Year\_Dummy_{it} + \eta_i + \mu_{it} \tag{2}$$

To examine the presence of a moderating role of CGQ in the relationship between IRQ disclosure and ICC, we introduce an interaction term in the fixed effect model in Eq. (3):

$$ICC_{it} = \beta_0 + \beta_1 IRQ_{it} + \beta_2 CGQ_{it} + \beta_3 (IRQ_{it} \times CGQ_{it}) + \beta_4 SIZE_{it} + \beta_5 LTG_{it} + \beta_6 ROA_{it} + \beta_7 CSR_{it}$$
$$+ \beta_8 LEV_{it} + \sum_{i=1}^{n} \beta_9 Company's\_Dummy_{it} + \sum_{i=1}^{n} \beta_{10} Year\_Dummy_{it} + \eta_i + \mu_{it} \tag{3}$$

In the regression model, ICC represents the implied cost of equity capital, IRQ denotes integrated reporting quality disclosure, CGQ signifies corporate governance quality, SIZE stands for company size, LTG represents long-term growth, ROA indicates return on assets, CSR denotes corporate social responsibility, LEV signifies leverage, $\beta_0$ is the intercept, $\beta_i$ represents the coefficient vector of explanatory variables, i denotes the i-th firm, t denotes the period, and $\eta$ and $\mu$ represent error terms. The data were analyzed using StataMP 16.0.

**3.2.5 Validity of empirical settings.** Typically, to manage multicollinearity concerns, researchers assess the variance inflation factors (VIF), ideally aiming for values below 10, as recommended by Hair et al. [134] and Gujarati [135]. In our investigation, VIF calculations indicated all variables maintained low values (below 5), signifying an absence of multicollinearity issues. The mean VIF, calculated at 3.83, substantiates data suitability for analysis, aligning with the established threshold.

Heteroscedasticity denotes the uneven distribution of error variances among observations, leading to "heteroskedastic" residual variance. Methods to detect this phenomenon include tests for constancy of error variance, such as Cameron and Trivedi's IM-test decomposition [136] and White's General Heteroscedasticity test [137]. In this study, the White test was employed (Table 1), revealing rejection of the null hypothesis (H$_0$), indicating heteroscedasticity in the model ($\chi^2 =$ 3.3e+05, p<0.001). To mitigate this, robust standard errors were applied to the model.

In this study, the uniformity of market capitalization among firms in Australia and New Zealand is presumed. However, to validate this assumption, commonly referred to as "homogeneity conditions" by Dyson et al. [138], a Levene test for homogeneity of variance was conducted [139]. The outcomes of the Levene F-test affirmed the assumption of variance homogeneity, yielding an F-value of (5.61) = 0.21 (p<0.04). Furthermore, the independent sample t-test exhibited a statistically significant impact, with t (520) = 2.41 (p<0.03). These findings demonstrated statistical significance in both the Levene and independent sample t-tests, thereby confirming the assumption of equal variance.

To address potential endogeneity concerns, we employed the Instrumental Variable Two-Stage Least Squares (IV2SLS) regression as a robustness check, given its capacity to manage endogeneity effectively. In our analysis, we identify possible endogeneity in IRQ, CGQ, and ROA. Following the recommendations of Baum et al. [140,141], IV2SLS requires valid instruments, and we used corporate governance reforms, environmental regulation changes, and firm age

**Table 1. Heteroskedasticity test.**

| Regression | Wald $\chi^2$ test | | |
|---|---|---|---|
| | $\chi^2$ (174) | Prob > $\chi^2$ | Null hypothesis, $H_0$ |
| $ICC_{it}$ | 3.3e+05 | 0.001 | Rejected |

as instruments for IRQ, CGQ, and ROA, respectively. The selection of instruments is based on their theoretical relevance and exogeneity. For corporate governance reforms, we included two significant policy events: the Australian Securities Exchange (ASX) Corporate Governance Principles and Recommendations (4th Edition) released in February 2019, which emphasized enhanced ESG disclosures to promote long-term value creation [142], and the New Zealand Exchange (NZX) ESG Guidance Note update in January 2020, which, though voluntary, spurred widespread ESG adoption among listed firms [143]. For environmental regulation changes, we relied on New Zealand's Climate Change Response (Zero Carbon) Amendment Act 2019, which established a framework for climate policies [144], and Australia's Climate Change Act 2022, legislating significant emission reduction targets [145]. Firm age was used as an instrument for ROA due to its strong correlation with operational efficiency and profitability (relevance) and its exogenous nature as a historical characteristic that impacts ICC only through ROA [146]. To validate the IV2SLS results, we conduct weak identification, under-identification, and overidentification tests. The Kleibergen-Paap rk Wald F statistic (WID) checks for weak identification, while the Kleibergen-Paap rk LM statistic (KPL) tests under-identification. Overidentification is examined using the Hansen J test, and endogeneity is assessed with the Durbin-Wu-Hausman (DW) test. A significant p-value for the KPL indicates the absence of under-identification, while WID values exceeding Stock-Yogo critical thresholds confirm the absence of weak identification. Additionally, the insignificance of the J and DW test p-values supports the absence of overidentification and endogeneity, respectively [140,141].

In addition to IV2SLS, we addressed endogeneity concerns in our baseline model using a two-step generalized method of moments (GMM) approach. This system GMM model, comprising a system of two equations, addresses common issues such as omitted variable bias, multicollinearity, and measurement errors inherent in pooled ordinary least squares (OLS) and fixed-effect regression models [147,148]. The effectiveness of the system GMM hinges on the integrity of its instruments and the absence of second-order serial correlation in the first-differenced residuals. We evaluated the validity of instruments using Hansen [149] over-identification test, which assesses the correlation between identified instruments and residuals. Acceptance of the null hypothesis indicates the appropriateness of instruments and the robustness of estimates, while rejection suggests their unreliability. Additionally, we examined second-order serial correlation in residuals through Arellano–Bond tests.

To further ensure robustness, we employed medium quantile regression, given its ability to handle outliers effectively and reduce sensitivity to extreme values. Additionally, pooled OLS was used as supplementary analyses to verify the consistency of findings and validate results obtained from fixed-effects regression models. Together, these approaches ensure the robustness and reliability of our results across various econometric techniques.

## 4. Results and discussion

### 4.1 Descriptive statistics

Table 2 displays the mean ICC for both Australia and New Zealand, averaging at 0.274, with a range from 0.128 to 0.567. The standard deviation of 0.091 suggests that while most companies had ICC values close to the mean, a few exhibited significantly higher or lower implied costs of capital. Compared to US and South African listed companies, Australian and New Zealand companies exhibited a higher average ICC, standing at 0.110 and 0.137, respectively [34,125], indicating relatively higher capital costs. Moreover, the mean IRQ score was 0.662, ranging from 0.109 to 3.124, with a substantial standard deviation of 0.391, indicating notable variation among sampled companies. This finding is in line with the

**Table 2. Descriptive statistics of dependent, moderating, and control variables.**

| Variables | Group | Mean | Std. Dev. | Min. | Max. |
|---|---|---|---|---|---|
| ICC | Overall | 0.274 | 0.091 | 0.128 | 0.567 |
| | Between | – | 0.041 | 0.146 | 0.346 |
| | Within | – | 0.072 | 0.050 | 0.510 |
| IRQ | overall | 0.662 | 0.391 | 0.109 | 3.124 |
| | between | – | 0.259 | 0.231 | 1.563 |
| | Within | – | 0.410 | -0.141 | 2.312 |
| CGQ | Overall | 0.309 | 0.131 | 0.119 | 0.693 |
| | Between | – | 0.064 | 0.150 | 0.406 |
| | Within | – | 0.111 | 0.010 | 0.338 |
| SIZE | Overall | 2.723 | 0.763 | 1.011 | 6.107 |
| | Between | – | 0.826 | 1.059 | 4.571 |
| | Within | – | 0.640 | 0.402 | 5.258 |
| LTG | Overall | 0.223 | 0.145 | 0.012 | 0.686 |
| | Between | – | 0.059 | 0.057 | 0.362 |
| | Within | – | 0.129 | -0.079 | 0.621 |
| ROA | Overall | 0.334 | 0.156 | 0.003 | 0.554 |
| | Between | – | 0.134 | 0.045 | 0.545 |
| | Within | – | 0.112 | -0.023 | 0.647 |
| CSR | Overall | 0.121 | 0.068 | 0.011 | 0.389 |
| | Between | – | 0.056 | 0.023 | 0.296 |
| | Within | – | 0.069 | -0.110 | 0.399 |
| LEV | Overall | 0.159 | 0.036 | 0.011 | 0.538 |
| | Between | – | 0.045 | 0.075 | 0.325 |
| | Within | – | 0.049 | -0.029 | 0.349 |

Note: Std. Dev., Min., and Max. denotes standard deviation, minimum, and maximum values, respectively.

integrated disclosure score index, where the average integrated reporting score was 0.395, with a maximum score of 1.00 [42,150,151]. The average IRQ score observed in this study was found to be lower than the IRQ score reported in India [4] and South Africa [34]. This discrepancy is attributed to the utilization of a disclosure checklist employing a comprehensive framework comprising content elements and guiding principles. This observation is consistent with the guidelines outlined in King III [152] and Songini et al. [99], advocating for the active involvement of the board, particularly through the audit committee, in verifying the accuracy of information contained within reports. Additionally, the average CGQ score in Australia and New Zealand was 0.309, suggesting a robust corporate governance system similar to that of South Africa, where the CGQ score was found to be 0.726 [130]. These scores align with the recommendation of King III [152], emphasizing the role of the board, particularly the audit committee, in ensuring report accuracy [12,153]. Pavlopoulos et al. [42] reported an average score of 0.538 in South Africa, reinforcing the importance of governance in integrated reporting accuracy. Table 2 presents descriptive statistics for the remaining control variables.

## 4.2 Correlation matrix among variables

Fig 3 illustrates the outcomes of the correlation analysis between the ICC and other associated variables. The results revealed a significant negative correlation of -0.104 between ICC and IRQ disclosure. This indicates firms with higher-quality disclosure tend to have lower ICC, which was consistent with prior research [33,69,154]. Similarly, ICC exhibited a negative correlation of -0.018 with CGQ, aligning with previous findings [80,82,119,155]. Moreover, IRQ disclosure

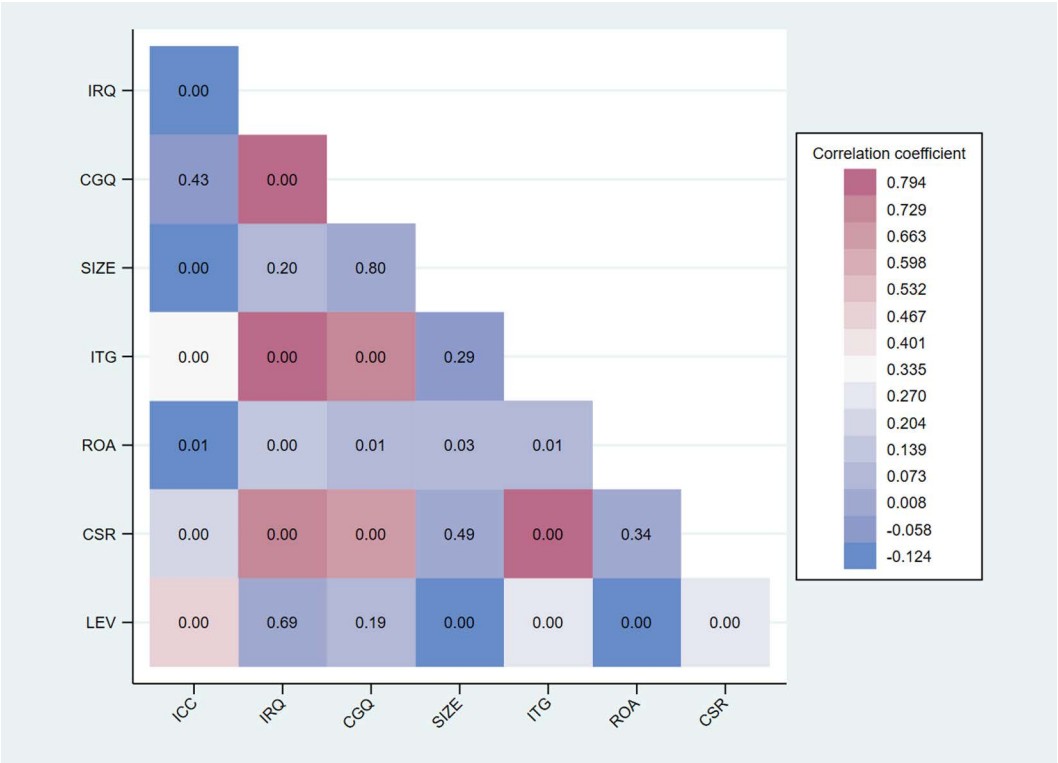

**Fig 3. The correlation coefficient of dependent, independent, and control variables.** The inner value displays the p-value corresponding to the correlation coefficient.

exhibited a notably positive and significant correlation of 0.693 with CGQ, that was consistent with existing literature [42,70,130].

In relation to other variables, ICC displayed a significant negative correlation of -0.164 with SIZE, while IRQ disclosure showed a positive yet insignificant correlation of 0.049 with SIZE. Furthermore, ICC demonstrated a significant positive correlation of 0.325 with LTG. However, ICC exhibited a significant negative correlation of -0.095 with ROA, while LTG displayed a significant positive correlation of 0.821 with IRQ disclosure. Additionally, ICC showed a significant positive correlation of 0.212 CSR, whereas CSR displayed a significant positive correlation of 0.702 with IRQ disclosure. Lastly, the results indicated a significant positive correlation of 0.454 between ICC and LEV. Overall, these findings supported the first hypothesis of the study and were consistent with prior research [44,119,154].

### 4.3 Moderating role of CGQ on IRQ disclosure and ICC

Table 3 presents the results of the fixed-effects regression models examining the moderating effect of CGQ on the relationship between IRQ disclosure and ICC. The analysis is conducted incrementally across three models to explore the direct effects, interaction effects, and the influence of additional control variables.

In Model I, which included IRQ and CGQ as the main explanatory variables, both coefficients were statistically significant and negative. Specifically, IRQ exhibited a coefficient of -0.238 ($p < 0.10$), indicating that higher-quality integrated reporting was associated with a reduction in ICC. Similarly, CGQ showed a coefficient of -0.079 ($p < 0.05$), suggesting that stronger corporate governance practices independently contributed to lowering the ICC. These findings highlighted the individual roles of IRQ and CGQ in mitigating capital costs.

**Table 3. Estimated coefficient of the moderation effect of CGQ between IRQ disclosure and ICC using a fixed-effect regression model.**

| Particulars | Model I | Model II | Model III |
|---|---|---|---|
| IRQ | -0.238* (0.125) | -0.327*** (0.058) | -0.262*** (0.057) |
| CGQ | -0.079** (0.033) | -0.341*** (0.089) | -0.405*** (0.072) |
| IRQ*CGQ | – | 0.257*** (0.035) | 0.287*** (0.036) |
| SIZE | – | – | 0.003 (0.003) |
| LTG | – | – | 0.831*** (0.232) |
| ROA | – | – | -0.017 (0.011) |
| CSR | – | – | -0.169*** (0.047) |
| LEV | – | – | 0.339*** (0.056) |
| Company fixed-effects | Yes | Yes | Yes |
| Year fixed-effects | Yes | Yes | Yes |
| Constant | 0.256*** (0.069) | 0.336*** (0.077) | 0.315*** (0.069) |
| F-statistic | 135.09*** | 142.57*** | 151.05*** |
| R-squared | 0.69 | 0.75 | 0.83 |
| Hausman test (Chi-square) | – | – | 67.61*** |
| Number of observations | 870 | 870 | 870 |
| No. of firm | 174 | 174 | 174 |

Robust standard errors are in parentheses. *** $p < 0.01$, ** $p < 0.05$, * $p < 0.10$.

Model II incorporated the interaction term between IRQ and CGQ (IRQ×CGQ) to assess the moderating effect of CGQ on the relationship between IRQ and ICC. The interaction term was positive and statistically significant (0.257, $p < 0.01$), indicating that CGQ strengthened the association between IRQ and ICC. The direct effects of IRQ (-0.327, $p < 0.01$) and CGQ (-0.341, $p < 0.01$) remained negative and statistically significant, reinforcing their independent contributions to lowering ICC. These results suggested that firms with higher governance quality derived greater benefits from integrated reporting in reducing their cost of capital. This outcome aligns with previous research [35,36,42,70,130] demonstrating a robust link between governance practices' quality and IRQ disclosure. Besides, the finding suggests that the IRQ coefficient markedly differs from model I when corporate governance is not considered, leading to a reduction in the ICC. Overall, these results support the agency theory within the voluntary disclosure theory proposed by Jensen and Meckling [97]. Consistent with the second hypothesis, CGQ played a moderating role in strengthening the relationship between IRQ disclosure and ICC for top-listed firms in Australia and New Zealand from 2018 to 2022.

In Model III, additional control variables—SIZE, LTG, ROA, CSR, and LEV—were included to account for firm-specific characteristics. IRQ and CGQ retained their significant and negative effects on ICC, with coefficients of -0.262 ($p < 0.01$) and -0.405 ($p < 0.01$), respectively. This observation aligns with prior research [33–36], which has consistently shown a negative correlation between the quality of integrated reporting and the cost of capital, thereby supporting the voluntary theory within the framework of signaling theory. This result also suggests that companies characterized by higher corporate governance standards tend to experience diminished costs associated with equity capital. Hence, top-tier firms listed in Australia and New Zealand, boasting high-quality corporate governance practices, demonstrate reduced costs of equity capital. This conclusion resonates with earlier studies [39,40,54,80,82,87] emphasizing the role of corporate governance in mitigating firms' capital costs by bolstering investor confidence in receiving adequate returns on their investments [126].

The interaction term (IRQ × CGQ) also remained significant and positive (0.287, p < 0.01), further supporting the moderating role of CGQ. This finding suggests that higher levels of corporate governance quality enhance the influence of integrated reporting quality on reducing the implied cost of equity capital. Such an outcome underscores the importance of robust governance structures in amplifying the benefits of integrated reporting practices, which aligns with recent research emphasizing the synergy between governance mechanisms and disclosure practices [39,40,42,70,130]. These results were consistent with the agency theory framework, which posits that high-quality governance reduces agency conflicts and fosters transparency, thereby mitigating information asymmetry [97]. Improved transparency through integrated reporting and effective governance structures increases investor confidence, as investors perceive reduced risks and greater assurance of adequate returns [33,34,36]. This finding also resonates with signaling theory, as the interaction effect suggests that firms with superior governance and reporting practices signal greater accountability and reliability to external stakeholders, thus reducing capital costs [40,80,87].

Recent empirical studies further substantiate this perspective. For instance, Chen and Zhang [156] demonstrate that corporate governance enhances the effectiveness of voluntary disclosure by improving stakeholder perceptions of accountability and risk management. Similarly, Zaid et al. [77] find that governance quality strengthens the relationship between ESG disclosures and financial outcomes, which aligns with the moderating effect observed in the present study. Furthermore, Gupta et al. [54] emphasize the role of governance quality in ensuring that disclosure practices are credible and impactful, thereby maximizing their influence on financial performance.

This study's findings advance the existing literature by empirically validating the interactive effect of IRQ and CGQ on ICC in the context of leading firms in Australia and New Zealand. The significant interaction term highlights the practical implications for firms and policymakers, indicating that combining high-quality governance with comprehensive integrated reporting can significantly reduce equity capital costs. This insight is particularly relevant in today's regulatory environment, where stakeholder demands for transparency and accountability continue to grow [14]. Future research could explore these dynamics in emerging markets, where governance standards and reporting practices may vary substantially, thereby offering a more nuanced understanding of their global implications.

Among the control variables, LTG demonstrated a robust positive relationship with ICC (0.831, p < 0.01), suggesting that firms with greater long-term growth opportunities faced higher implied costs of equity capital. This finding supported prior research on the influence of growth prospects on equity costs [112,157]. Conversely, CSR practices were negatively associated with ICC (-0.169, p < 0.01), highlighting that socially responsible practices enhanced firm reputation and lowered perceived risks [44,156]. Leverage exhibited a positive relationship with ICC (0.339, p < 0.01), consistent with the notion that higher financial risk increased investors' required returns [156,158,159]. However, company's size and return on assets did not show statistically significant effects, indicating that these factors may have had limited influence on ICC in the given context.

Regarding model fit, the R-squared values across the models progressively increase from 0.69 in Model I to 0.83 in Model III, indicating improved model fit with the inclusion of interaction terms and control variables. Furthermore, the Hausman test statistic in Model III ($\chi^2$ = 67.61, p < 0.01) confirms the appropriateness of the fixed-effects regression specification, ensuring the reliability of the estimates.

To delve deeper into the intricacies of moderations, we adhered to the guidelines outlined by Cohen et al. [160] and conducted a simple slope analysis, as recommended by Aiken and West [161]. The slope analysis depicted in Fig 4 illustrates how CGQ moderates the relationship between IRQ disclosure and ICC. The analysis revealed that the relationship between IRQ disclosure and ICC varied significantly depending on the level of CGQ. Specifically, under weak CGQ, there was a steeper negative slope between IRQ disclosure and ICC, indicating that as IRQ disclosure increased, ICC decreased substantially. In contrast, under strong CGQ, the negative association was less pronounced, resulting in a flatter slope. The rationale for weaker CGQ implying a lower ICC, as illustrated in Fig 4, lay in the compensatory role played by high-quality integrated reporting. Firms with weaker governance standards often faced greater investor skepticism due

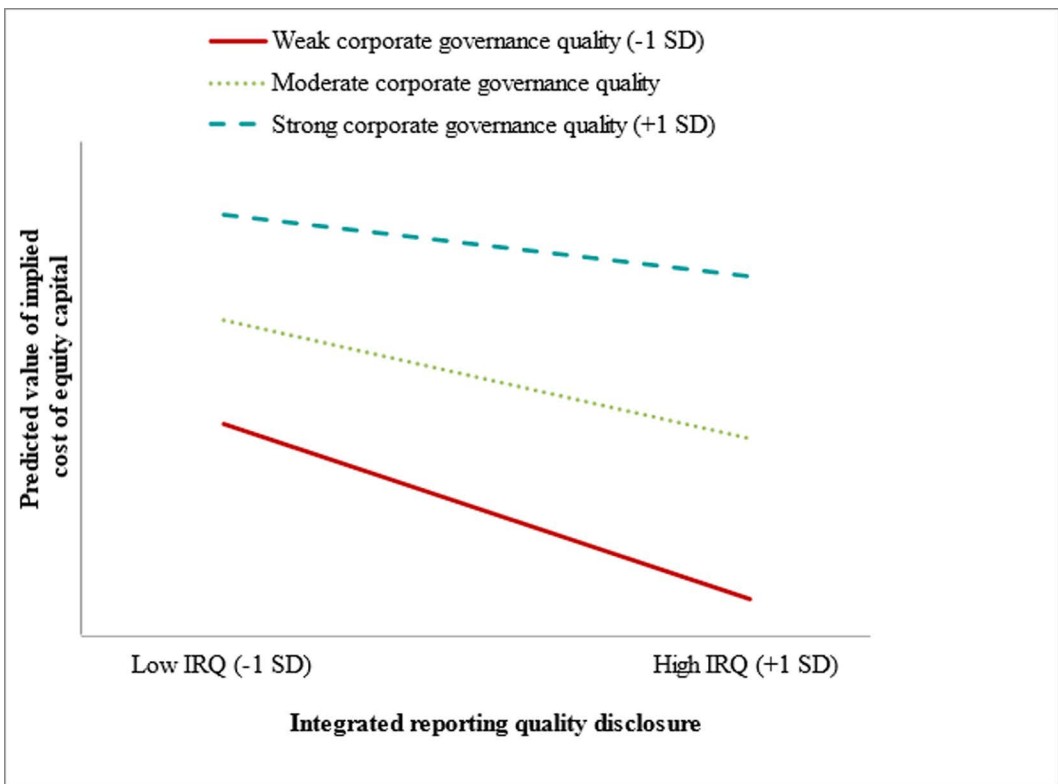

**Fig 4. A conditional margin plot for moderating effect of CGQ on the relationship between IRQ disclosure and ICC.**

to higher perceived risks, such as agency conflicts or managerial opportunism [97]. In such cases, high-quality IRQ disclosures served as a compensatory mechanism, signaling transparency and accountability to investors [36], thereby reducing ICC more effectively [44,156]. This compensatory effect diminished when CGQ was strong because robust governance structures already instilled confidence in investors, leading to relatively smaller incremental benefits from improved IRQ disclosure [34,54].

Furthermore, Fig 4 indicated that the difference in ICC between low and high CGQ levels was more pronounced when IRQ disclosure was high. This suggested that high IRQ disclosure was particularly impactful for firms with weaker governance, as it directly addressed gaps in investor confidence and mitigated concerns related to information asymmetry. Conversely, when IRQ disclosure was low, the difference in ICC across CGQ levels was smaller, highlighting the limited efficacy of low IRQ disclosure in reducing ICC regardless of governance quality.

In conclusion, the regression results and Fig 4 collectively demonstrate that CGQ moderates the relationship between IRQ disclosure and ICC. The compensatory role of IRQ in weaker CGQ settings underscores its critical importance in improving investor perceptions and reducing ICC, particularly in developed markets.

In addition, adopting the approach outlined by Rast et al. [162], we employed an effective analytical tool known as the Johnson-Neyman technique, also referred to as 'floodlight analysis'. The aim was to identify and illustrate statistically significant regions, as indicated by the 95% confidence intervals around the basic slopes at various levels of the moderator—CGQ in this instance, as described by Hayes [163]. As shown in Fig 5, the graph clearly delineates the bounds of significance. The upper and lower limits of the confidence interval intersect the zero line of the vertical axis between CGQ values of 0.52 and 0.70, marking a threshold point revealed by this visual representation. At this juncture, the relationship

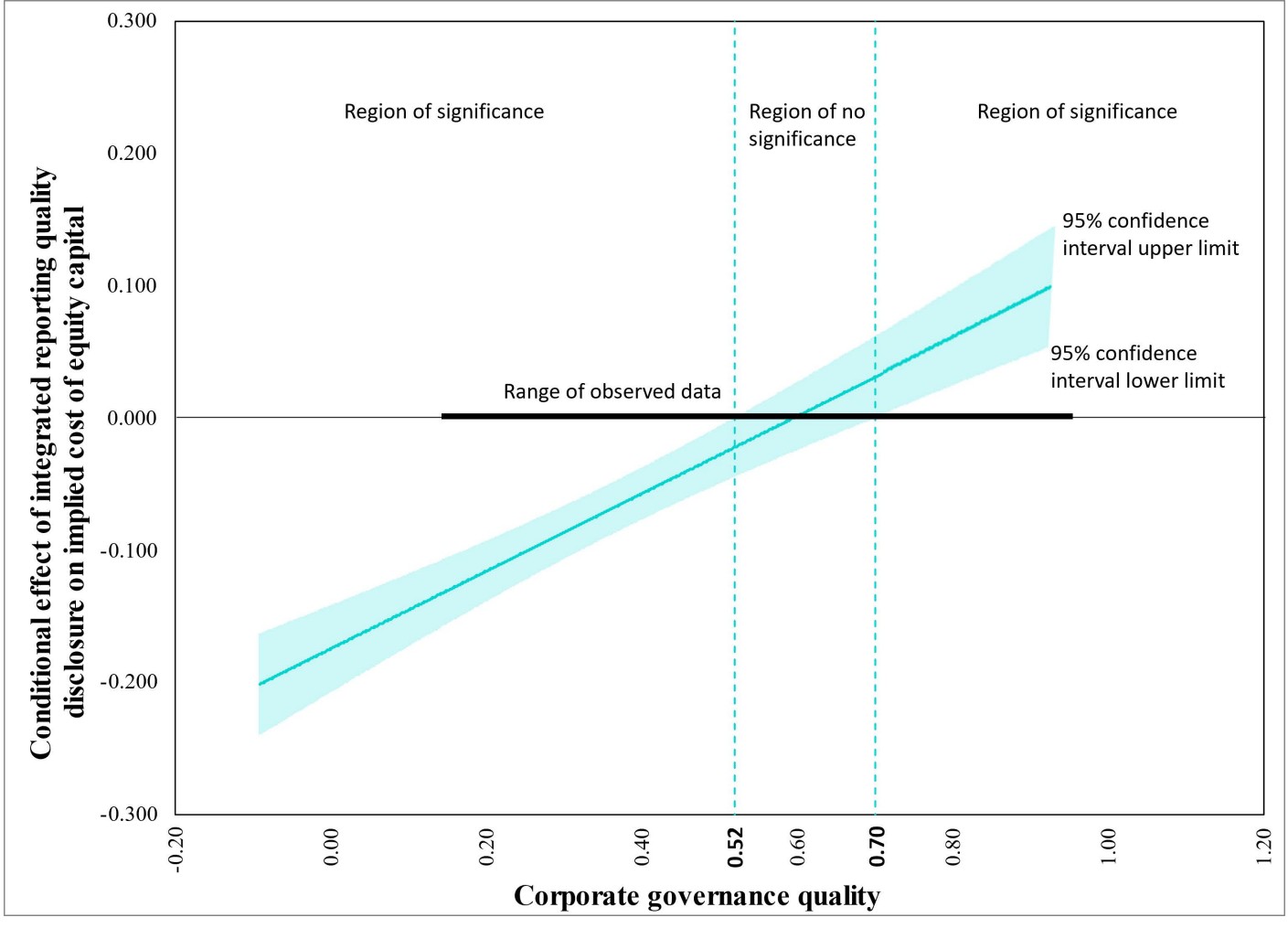

**Fig 5. Regions of significance for the relationship between IRQ disclosure and ICC at various levels of CGQ (Johnson-Neyman plot).**

between IRQ disclosure and ICC becomes statistically significant. Put succinctly, for all CGQ levels below 0.52 and above 0.70, the conditional effect of IRQ disclosure on ICC remained significant.

## 4.4 Robustness of the results

In our study, we assessed the robustness and reliability of the baseline model outcomes through the application of four distinct methodologies: Two-Step system GMM (Model 1), pooled OLS (Model 2), medium quantile regression (Model 3), and IV2SLS (Model 4). Each of these techniques was utilized with the dataset to ascertain the strength and coherence of the findings, as depicts in Table 4.

The Two-Step System GMM model was utilized to address potential endogeneity concerns and enhance the precision of estimations [147,148,164]. This approach effectively mitigates unobserved heterogeneity and inter-variable correlation, which could otherwise bias estimates in the baseline model. Using moment conditions derived from the independence of instrumental variables and error terms, the GMM model enables parameter estimation while tackling omitted variable bias and reverse causality. It is particularly suitable for dynamic panel data analysis as it incorporates both lagged and

**Table 4. Estimated coefficient of the two-step system GMM, pooled OLS, medium quantile regression, and IV2SLS models for robustness check.**

| Particulars | Model 1 | Model 2 | Model 3 | Model 4 |
|---|---|---|---|---|
| L.ICC | 0.072* (0.038) | – | – | – |
| IRQ | -0.284*** (0.087) | -0.323*** (0.069) | -0.295*** (0.044) | -0.196*** (0.055) |
| CGQ | -0.319** (0.131) | -0.372*** (0.114) | -0.306*** (0.096) | -0.205** (0.094) |
| IRQ*CGQ | 0.235** (0.111) | 0.293*** (0.045) | 0.258*** (0.082) | 0.173** (0.074) |
| SIZE | -0.004 (0.003) | -0.002 (0.002) | 0.003 (0.003) | -0.002 (0.012) |
| LTG | 0.824*** (0.253) | 0.879*** (0.157) | 0.801*** (0.143) | 0.601** (0.286) |
| ROA | 0.003 (0.012) | -0.003 (0.015) | -0.009 (0.011) | 0.011 (0.021) |
| CSR | -0.113*** (0.033) | -0.098** (0.039) | -0.124*** (0.033) | -0.082*** (0.021) |
| LEV | 0.226*** (0.065) | 0.253*** (0.061) | 0.239*** (0.042) | 0.219*** (0.048) |
| Constant | 0.238*** (0.038) | 0.259*** (0.057) | – | 0.345*** (0.053) |
| Observations | 696 | 870 | 870 | 870 |
| Number of groups/firm | 174 | 174 | 174 | 174 |
| Number of instruments | 13 | – | – | 3 |
| R-squared | – | 0.86 | – | 0.73 |
| F-statistic | – | 159.75*** | – | 118.03*** |
| Wald chi$^2$ | 4253.65*** | – | – | 19.53** |
| Arellano-Bond test for AR(1) in first differences, z | -4.02*** | – | – | – |
| Arellano-Bond test for AR(2) in first differences, z | -0.58 | – | – | – |
| Sargan test of overid. Restrictions, chi$^2$ (p-value) | 12.06 (0.231) | – | – | – |
| Hansen test of overid. Restrictions, chi$^2$ (p-value) | 8.18 (0.203) | – | – | – |
| Difference-in-Hansen tests of exogeneity of instrument subsets: GMM instruments for levels | – | – | – | – |
| Hansen test excluding group, chi$^2$ (p-value) | 6.12 (0.201) | – | – | – |
| Difference (null H = exogenous), chi$^2$ (p-value) | 2.46 (0.328) | – | – | – |
| Instrument variable context | – | – | – | – |
| Hansen test excluding group, chi$^2$ (p-value) | 4.98 (0.411) | – | – | – |
| Difference (null H = exogenous), chi$^2$ (p-value) | 4.81 (0.178) | – | – | – |
| KPL (p-value) | – | – | – | 87.54 (0.001) |
| WID | – | – | – | 161.85 |
| J (p-value) | – | – | – | 0.152 (0.169) |
| DW (p-value) | – | – | – | 2.383 (0.223) |

Note: Models 1, 2, 3, and 4 denotes two-step system GMM, pooled OLS, medium quantile regression, and instrumental variable two-stage least square (IV2SLS) regression, respectively. Endogenous variables of two-step system GMM model: Implied cost of equity capital (ICC), integrated reporting quality (IRQ) disclosure, corporate governance quality (CGQ), and interaction between IRQ disclosure and CGQ. Instrumental variables of two-step system GMM model: lag of the endogenous variables. Instrumental variables of IV2SLS model: corporate governance reforms, environmental regulation changes, and firm size. WID = The Kleibergen-Paap rk Wald F (weak identification) test statistic. KPL = Kleibergen-Paap rk LM (under-identification) test statistic. J = Hansen over-identification test statistic. DW = Durbin-Wu-Hausman endogeneity test statistic. Robust standard errors in parentheses. *** $p < 0.01$, ** $p < 0.05$, * $p < 0.10$.

contemporaneous variables [148]. In our dataset, variables such as ICC, IRQ disclosure, CGQ, and their interactions were treated as endogenous, and instrumental variables derived from lagged values were employed to address endogeneity. The validity of this approach hinges on the assumption that lagged values are uncorrelated with the current error term [148].

In our investigation, the Arellano-Bond test was employed to detect any presence of autocorrelation in the error term within the first-differenced model. The analysis revealed significant first-order autocorrelation (z = -4.02, p < 0.01) but no

evidence of second-order autocorrelation (z = -0.58, p > 0.10). This indicates that while there is correlation between current and past error terms, it does not compromise the model's validity. Instrument validity was tested using the Sargan and Hansen tests, which yielded p-values of 0.231 and 0.203, respectively, confirming that the instruments were not correlated with the error term and were jointly valid. A significant Wald chi-squared value ($\chi^2 = 4253.65$, p < 0.01) further underscored the statistical significance of the regression coefficients, affirming the predictive accuracy of the model.

The system-GMM results indicated that the lag of ICC had a significant positive effect on current ICC (0.072, p < 0.10). Both IRQ disclosure and CGQ had significant negative effects on ICC (-0.284, p < 0.01 and -0.319, p < 0.05, respectively). However, their interaction demonstrated a significant positive effect on ICC (0.235, p < 0.05). LTG and leverage positively influenced ICC (0.824, p < 0.01 and 0.226, p < 0.01, respectively), while CSR showed a significant negative impact (-0.113, p < 0.01). Company size and ROA were not significant predictors. These findings emphasize the importance of IRQ disclosure, CGQ, LTG, CSR, and leverage as determinants of ICC. Furthermore, the consistency between system-GMM results and the baseline fixed-effects model bolsters the robustness of the conclusions.

The Pooled OLS model provided additional evidence supporting the study's findings (2nd model of Table 4). Its simplicity and interpretability rendered it an effective tool for examining variable relationships. The regression coefficients were consistent with those of the main model, confirming the stability of results. The third model in Table 4 showed that the regression coefficients had identical signs and significance levels as those observed in the fixed-effects model (Table 3). This consistency highlights the robustness of the findings across quantiles, particularly in addressing heterogeneity within the dataset.

The IV2SLS model (fourth model in Table 4) confirmed the absence of under-identification through the Kleibergen-Paap rk LM statistic (p < 0.001). The weak identification test indicated strong instruments, and the Hansen J statistic (p = 0.169) affirmed their joint validity. The Durbin-Wu-Hausman test confirmed the absence of endogeneity (p = 0.223). In this model, IRQ disclosure and CGQ had significant negative effects on ICC (-0.196, p < 0.01 and -0.205, p < 0.05, respectively), while their interaction showed a positive effect (0.173, p < 0.05). LTG and leverage exhibited significant positive effects (0.601, p < 0.05 and 0.219, p < 0.01, respectively), whereas CSR negatively impacted ICC (-0.082, p < 0.01). Company size and ROA did not have significant effects. These results were consistent with the fixed-effects model, reinforcing the reliability and robustness of the study's conclusions.

## 5. Conclusions and way forward

This study examined the impact of integrated reporting quality disclosure on the implied cost of equity capital and explored the moderating role of corporate governance quality. Focusing on Australia and New Zealand, two developed markets with advanced regulatory frameworks, the findings reveal a significant negative relationship between IRQ disclosure and ICC, demonstrating that firms adopting higher-quality integrated reporting enjoy lower capital costs. Furthermore, CGQ was found to enhance the relationship between IRQ and ICC, underscoring the complementary role of robust governance structures in amplifying the benefits of integrated reporting. These findings advance existing literature by providing new insights into the interplay between reporting quality, governance, and financial outcomes in developed markets.

The study makes three key contributions to the literature. First, it extends the understanding of voluntary information disclosure by highlighting the joint impact of IRQ and CGQ on ICC, particularly in the context of mature markets. Second, it employs advanced econometric methods, such as IV2SLS and system-GMM, to address potential endogeneity issues, setting a methodological benchmark for future research. Third, it provides practical insights for firms and policymakers, emphasizing that adopting high-quality integrated reporting and enhancing governance standards can lower financing costs and improve investor confidence. By refining agency theory, the study demonstrates that integrated reporting mitigates information asymmetry between firms and investors, while strong governance structures reduce agency conflicts, further lowering the cost of capital.

These findings carry significant implications for firms in jurisdictions with similar governance frameworks and market dynamics. Firms contemplating the adoption of integrated reporting can leverage these insights to strengthen their disclosure practices, enhance transparency, and attract cost-effective capital. However, the study's findings may not be

universally applicable to markets with differing governance or regulatory environments, warranting further research to generalize these results.

Despite its contributions, this study has certain limitations. The focus on firms in Australia and New Zealand restricts the geographical scope of the findings. Additionally, the study does not establish causal relationships between IRQ and ICC, highlighting the need for future research employing causal inference techniques, such as machine learning methods. Further investigations could explore the effects of alternative disclosure practices, such as sustainability or environmental reporting, on the cost of equity capital. Future studies could also examine the implications of IRQ adoption on firm performance, shareholder value, and executive decision-making, while categorizing firms based on the extent of their IRQ disclosure (full, partial, or non-adoption).

In conclusion, this study enriches the literature on voluntary information disclosure by providing robust evidence of the financial benefits of integrated reporting and corporate governance quality. By bridging gaps in the existing research, it offers valuable guidance for academics, practitioners, and policymakers seeking to optimize reporting and governance practices in a global context.

## Author contributions

**Conceptualization:** Muhammad Arslan Iqbal, Md. Abdur Rouf Sarkar.

**Data curation:** Muhammad Arslan Iqbal, Md. Abdur Rouf Sarkar, Md Jahid Ebn Jalal.

**Formal analysis:** Muhammad Arslan Iqbal, Md. Abdur Rouf Sarkar, Md Jahid Ebn Jalal.

**Investigation:** Md. Abdur Rouf Sarkar, Majed Alharthi.

**Methodology:** Muhammad Arslan Iqbal, Md. Abdur Rouf Sarkar, Majed Alharthi, Md. Naimur Rahman.

**Resources:** Md. Abdur Rouf Sarkar, Md. Naimur Rahman.

**Software:** Muhammad Arslan Iqbal.

**Supervision:** Majed Alharthi.

**Validation:** Majed Alharthi, Md Jahid Ebn Jalal.

**Visualization:** Muhammad Arslan Iqbal, Md. Abdur Rouf Sarkar.

**Writing – original draft:** Muhammad Arslan Iqbal, Md. Abdur Rouf Sarkar.

**Writing – review & editing:** Majed Alharthi, Md. Naimur Rahman.

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
