## [Decision Letter · Decision Letter 0]

23 Jul 2024

PONE-D-24-26051Effect of integrated reporting quality disclosure on cost of equity capital in developed markets: Exploring the moderating role of corporate governance qualityPLOS ONE

Dear Dr. Sarkar,

Thank you for submitting your manuscript to PLOS ONE. After careful consideration, we feel that it has merit but does not fully meet PLOS ONE’s publication criteria as it currently stands. Therefore, we invite you to submit a revised version of the manuscript that addresses the points raised during the review process.

We look forward to receiving your revised manuscript.

Kind regards,

Yongliang Yang, Ph.D.

Academic Editor

PLOS ONE

Journal Requirements:

2. In this instance it seems there may be acceptable restrictions in place that prevent the public sharing of your minimal data. However, in line with our goal of ensuring long-term data availability to all interested researchers, PLOS’ Data Policy states that authors cannot be the sole named individuals responsible for ensuring data access (http://journals.plos.org/plosone/s/data-availability#loc-acceptable-data-sharing-methods).

**Additional Editor Comments:**

The topic of this article is very important, but it is also a classic topic, so there is a wealth of relevant research literature. The author needs to better extract the innovative points of this article in the introduction and expand the inspiration of this article for existing research in the conclusion.

When estimating the IRQ on cost of equity capital in this article, the identification strategy needs to be more refined. Currently, it is difficult to avoid potential endogeneity issues, and some robustness results need to be supplemented.

This article refers to many references, but the summary of relevant literature in the past two years is insufficient, making it difficult to reflect the current research progress and trends. It is recommended that the author make structural adjustments in terms of references.

Reviewers' comments:

Reviewer's Responses to Questions

**Comments to the Author**

1. Is the manuscript technically sound, and do the data support the conclusions?

Reviewer #1: Yes

Reviewer #2: Partly

2. Has the statistical analysis been performed appropriately and rigorously? 

Reviewer #1: Yes

Reviewer #2: No

3. Have the authors made all data underlying the findings in their manuscript fully available?

Reviewer #1: Yes

Reviewer #2: No

4. Is the manuscript presented in an intelligible fashion and written in standard English?

Reviewer #1: Yes

Reviewer #2: Yes

5. Review Comments to the Author

Reviewer #1: This article analyzes a topical issue: the relationship between the quality of integrated reporting (IRQ) and the cost of capital, including the effect of corporate governance quality (CGQ). Authors rely on agency theory and argue that to reduce information asymmetry, firms issue more information which allows to reduce the cost of capital. Therefore, a more comprehensive firms’ reporting, including information about their strategy and risks, increases investors’ confidence, hence it may reduce the cost of capital. According authors their contribution consists of analyzing the role of information about employees.

The structure of the article is correctly chosen and it exhibits a logical concern. However, I present some comments:

i. In introduction authors mentioned stakeholders and shareholders on line 66. I suggest to remove shareholder or including “in particular” because shareholders are stakeholders too.

ii. Regarding literature review, it is not clear what authors add to previous literature given that the relation between IRQ and Cost of Capital and the influence of CGQ has been studied before, for example De Villiers et al. (2017), Pavlopoulos et al. (2017) and Gupta et al. (2018).

iii. Authors use a unique variable to estimate CGQ which may imply loosing relevant information about different mechanism of corporate governance. It would be beneficial to support their argument using other studies showing that it brings transparency and captures stakeholders’ confidence. In addition, the definition of the control variable size should be presented in the text (market capitalization, ln total assets, or other) as well as the definition of Lev.

iv. In equation 2, presented on line 460, the interaction term is already included, as well as the control variables, which reduces the usefulness of equation 3 and 4.

iv. Regarding results documented on table 3, authors do not comment the coefficient of interaction term IRQ and CGQ, which is statistically significant at 1% level, but positive, mitigating the reduction of the cost of capital. This result is just mention after, when authors analyze table 4. In any case, an economic argument must be provided for this result.

In addition, I suggest to remove from the legend the p values that are not presented in the table (* and **)

v. Once again, it would be relevant having the rational for weaker CGG implying lower ICC for each IRQ, as illustrated in figure 4.

vi. In line 630, it would be clearer to mention that in table 5, model 1 corresponds to GMM, model 2 to pooled OLS and model 3 to mean quantile regression.

Reviewer #2: First, please provide more detailed descriptions of data sources and processing procedures: Ensure the reliability of the data and the transparency of the methodology.

Second, please explain why the firm-level fixed effect is not included in the models, and enhance the detail and explanation in the results section.

6. PLOS authors have the option to publish the peer review history of their article (what does this mean? ). If published, this will include your full peer review and any attached files.

**Do you want your identity to be public for this peer review?** For information about this choice, including consent withdrawal, please see our Privacy Policy .

Reviewer #1: No

Reviewer #2: No

---

## [Author Response · Author response to Decision Letter 1]

23 Jan 2025

Manuscript number: PONE-D-24-26051

Title: Effect of integrated reporting quality disclosure on cost of equity capital in developed markets: Exploring the moderating role of corporate governance quality

Journal: PLOS ONE

Thank you for the comments concerning our manuscript. We deeply appreciate your positive evaluation of our work. Those comments are valuable and very helpful. We have read through comments carefully and have made corrections. Please see below; all tasks and revisions taken are shown point-by-point.

Response to Academic Editor comments

Comments #1: Please ensure that your manuscript meets PLOS ONE's style requirements, including those for file naming.

Response to comment #1: Thank you for your comment. We have carefully reviewed and ensured that the manuscript now fully complies with PLOS ONE's style requirements, including the file naming conventions. Please let us know if there are any additional adjustments required.

Comments #2: In this instance it seems there may be acceptable restrictions in place that prevent the public sharing of your minimal data. However, in line with our goal of ensuring long-term data availability to all interested researchers, PLOS’ Data Policy states that authors cannot be the sole named individuals responsible for ensuring data access (http://journals.plos.org/plosone/s/data-availability#loc-acceptable-data-sharing-methods).

Response to comment #2: Thank you for your comment. We have uploaded a minimum data as a supplementary file.

Comments #3: The topic of this article is very important, but it is also a classic topic, so there is a wealth of relevant research literature. The author needs to better extract the innovative points of this article in the introduction and expand the inspiration of this article for existing research in the conclusion.

Response to comment #3: Thank you for your thoughtful and constructive comment. We have revised the introduction and conclusion sections to clearly articulate the innovative contributions of this study and to expand on its implications for existing research. Additionally, we have updated the literature review sections (2.1 and 2.2) to provide a detailed discussion of the research gap this study addresses. These revisions aim to better highlight the unique aspects of our work and its relevance to the existing body of literature. Please refer to the revised Sections 1, 2.1, 2.2, and 5 for these updates.

Comments #4: When estimating the IRQ on cost of equity capital in this article, the identification strategy needs to be more refined. Currently, it is difficult to avoid potential endogeneity issues, and some robustness results need to be supplemented.

Response to comment #4: Thank you for your insightful comment. To address the potential endogeneity issues, we have incorporated two econometric models—IV2SLS and system-GMM—under the "Robustness of the Results" section. These models were employed to ensure the reliability of our findings and mitigate endogeneity concerns. Please refer to the revised text in Sections 3.2.5 and 4.4 for further details.

Comments #5: This article refers to many references, but the summary of relevant literature in the past two years is insufficient, making it difficult to reflect the current research progress and trends. It is recommended that the author make structural adjustments in terms of references.

Response to comment #5: Thank you for your valuable suggestion. We have included the latest references relevant to our research context. Please refer to the revised manuscript for further details.

Response to Reviewer #1 comments

Comments #1: This article analyzes a topical issue: the relationship between the quality of integrated reporting (IRQ) and the cost of capital, including the effect of corporate governance quality (CGQ). Authors rely on agency theory and argue that to reduce information asymmetry, firms issue more information which allows to reduce the cost of capital. Therefore, a more comprehensive firms’ reporting, including information about their strategy and risks, increases investors’ confidence, hence it may reduce the cost of capital. According authors their contribution consists of analyzing the role of information about employees. The structure of the article is correctly chosen and it exhibits a logical concern.

Response to comment #1: We sincerely thank the reviewer for the thoughtful and encouraging feedback on our manuscript. We appreciate your recognition of the topical nature of our study and its reliance on agency theory to examine the interplay between integrated reporting quality, corporate governance quality, and the cost of capital. Your acknowledgment of the logical structure and clear exposition of our argument is particularly gratifying. Additionally, we are pleased that you noted our emphasis on the role of employee-related information in our analysis. We believe this aspect adds a unique dimension to understanding how integrated reporting enhances transparency and reduces information asymmetry, ultimately contributing to reducing the cost of capital. Thank you once again for your constructive comments, which is undoubtedly strengthen the quality of our work.

Comments #2: In introduction authors mentioned stakeholders and shareholders on line 66. I suggest to remove shareholder or including “in particular” because shareholders are stakeholders too.

Response to comment #2: We thank the reviewer for the insightful comment regarding the phrasing on line 66. We have revised the sentence to clarify the relationship between stakeholders and shareholders, acknowledging that shareholders are a subset of stakeholders. The revised sentence now reads: "Stakeholders, particularly shareholders, now regularly demand comprehensive financial information."

Comments #3: Regarding literature review, it is not clear what authors add to previous literature given that the relation between IRQ and Cost of Capital and the influence of CGQ has been studied before, for example De Villiers et al. (2017), Pavlopoulos et al. (2017) and Gupta et al. (2018).

Response to comment #3: Thank you for your valuable feedback. We have revised Sections 2.1 and 2.2 of the literature review to incorporate the research gap addressed in our manuscript. The updated sections now provide a clearer articulation of the gap and its significance to our study. Kindly refer to the revised Sections 2.1 and 2.2, as detailed in the text below:

“2.1 Relationship between IRQ disclosure and ICC

Integrated reporting has gained attention as a mechanism for improving financial transparency and aligning corporate governance with stakeholder expectations (de Villiers, Venter and Hsiao, 2017; Pavlopoulos, Magnis and Iatridis, 2017). By integrating financial and non-financial disclosures, IR aims to enhance decision-making, reduce information asymmetry, and ultimately lower the cost of capital. Non-financial data, including CSR initiatives, have been observed to correlate with the cost of equity capital. Breuer et al. (2018) investigated the link between CSR, investor protection, and equity capital costs, noting that in jurisdictions with robust investor safeguards, CSR investment tends to lower equity costs, whereas in contexts with weaker investor protection, such investments tend to elevate equity costs.

Theoretical frameworks, as elucidated by Dhaliwal et al. (2011), underscore the significance of employing prospective information to gauge equity capital costs. Voluntary disclosure of non-financial data is viewed as a means to bridge the gap between external and internal information, thereby diminishing equity capital costs and attracting long-term investors. Moreover, following the mandatory adoption of international financial reporting standards (IFRS), there was an enhancement in accounting information quality (Barth, Landsman and Lang, 2008; Armstrong, Guay and Weber, 2010; Devalle, Onali and Magarini, 2010), leading to a subsequent reduction in capital costs (Lee and Yeo, 2010; LI, 2010; Hoque, 2017). Enhanced information quality is associated with fewer instances of managerial misappropriation, thereby resulting in lower equity capital costs (Zhou, 2014). Nonetheless, certain studies have identified a negative correlation between voluntary non-financial information disclosure and equity capital costs (Griffin and Sun, 2013; Plumlee et al., 2015).

Integrated reporting represents a voluntary reporting framework that holds promise for revolutionizing corporate reporting practices (Williams and Lodhia, 2021). Recent data from a survey conducted by PricewaterhouseCoopers (PwC) reveals that approximately two-thirds of investment professionals perceive the quality of a company's reporting, encompassing details about strategy, risks, and other value drivers, as directly influencing the cost of capital (Songini et al., 2023). Nonetheless, a dearth of studies exists investigating the nexus between IRQ and the capital market, warranting further exploration in this domain to enhance comprehension of integrated reporting's impacts on the cost of equity capital.

Prior studies have explored the relationship between the quality of financial disclosures and a firm’s cost of capital (Gupta, Krishnamurti and Tourani-Rad, 2018). Research using global datasets analyzed IR and capital costs across 27 countries, covering 995 firms and 3,294 observations from 2009 to 2013 (García-Sánchez and Noguera-Gámez, 2017; Zhou, Simnett and Green, 2017). Maria and Ligia’s (2017) found a negative association between IRQ disclosure and the cost of capital but noted that differences in corporate governance systems among countries did not significantly influence this relationship. In contrast, Barth et al. (2017), studying 80 firms listed on the Johannesburg Stock Exchange, found no significant relationship between IRQ and ICC. Similarly, Lee and Yeo (2016) observed a positive association between IRQ disclosure and firm valuation, contributing to mixed findings on the IRQ-ICC relationship.

Existing literature also highlights limitations. For instance, while de Villiers et al. (2017) primarily focused on the conceptual underpinnings of IR, emphasizing its potential to provide a comprehensive narrative of value creation but lacking empirical insights into its impact on cost of capital. Pavlopoulos et al. (2017) identified a positive association between IR quality and corporate governance but did not explore how this interaction affects financial outcomes like cost of equity. Similarly, Gupta et al. (2018) demonstrated the significance of governance mechanisms and financial development in reducing cost of equity but did not examine IR as a specific governance tool. This leaves a critical gap in understanding how IRQ interacts with CGQ to influence a firm’s financial outcomes, particularly within different institutional contexts.

Our study aims to address this gap by investigating the direct and interactive effects of IRQ and CGQ on the cost of capital. Unlike prior research, our approach explicitly incorporates the dynamic interplay between these variables, offering insights into their complementary roles across diverse regulatory and financial environments. By doing so, this study contributes to a more nuanced understanding of the mechanisms through which IR impacts financial metrics, expanding the scope of existing literature.

2.2 Relationship among IRQ disclosure, CGQ, and ICC

Previous investigations have provided empirical support for the advantages associated with the adoption of high-quality integrated reporting (IIRC, 2013; de Villiers, Rinaldi and Unerman, 2014; Adrián-Martínez et al., 2016; Hoque, 2017). However, an exclusive focus on IRQ may not furnish a comprehensive understanding of its influence on capital market outcomes, particularly its efficacy on the cost of capital, without accounting for a robust institutional framework. Accounting practices, as evidenced in prior studies, were shaped by diverse factors, including the underlying capital markets (Walker, 2010; Wysocki, 2011; Cieslewicz, 2014; Wehrfritz and Haller, 2014). While certain studies have suggested a tenuous relationship between IRQ disclosure and ICC (Lee and Yeo, 2016; Barth et al., 2017; García-Sánchez and Noguera-Gámez, 2017; Zhou, Simnett and Green, 2017), there exists a correlation between an integrated reporting framework and the capital market (Zhou, 2014).

Nonetheless, extant research has predominantly concentrated on the impact of high-quality disclosure on the capital market, overlooking other institutional variables, despite theoretical assertions regarding the synergy between high-quality reporting disclosure and other institutional factors. Previous study has illustrated that CGQ has been employed as a moderating variable in numerous investigations (Chen and Paulraj, 2004; Alhazaimeh, Palaniappan and Almsafir, 2014; Ould Daoud Ellili, 2020; Abu Alia et al., 2022; Tawfik et al., 2022; Liu, Li and Lin, 2023).

Our research endeavors to bridge this void by exploring the influence of corporate governance on both the IRQ disclosure and ICC. Through the examination of employees' role in CGQ as a moderating variable, we aim to offer valuable insights into how governance practices shape the relationship between IRQ disclosure and ICC. In this segment, we delve into the correlation between IRQ disclosure and ICC (Section 2.1). Subsequently, we broaden our investigation to encompass the association between IRQ disclosure, CGQ, and ICC (Section 2.2). Previous studies have already scrutinized the nexus between voluntary disclosure, encompassing both financial and non-financial aspects, and corporate governance (Hail and Leuz, 2006; Fifka and Pobizhan, 2014; Ortas et al., 2015; Haldar and Raithatha, 2017).

A pioneering approach to enhancing CSR reporting is IR, which endeavors to amalgamate CSR disclosures (Sun et al., 2022). Furthermore, scholars have noted a positive linkage between CGQ and CSR, with Huang (2010) documenting a robust correlation. Similarly, in emerging economies, Jahid et al. (2022) have affirmed the influence of CGQ practices on CSR disclosures. However, Sriani and Agustia (2020) identified no significant association between IRQ and information asymmetry as measured by the spread. The ESG information is pivotal in evaluating companies' risk profiles, future cash flows, and effectively pricing investments (Veltri et al., 2023). Institutional factors, such as corporate governance, directly shape voluntary disclosure and its ramifications on equity costs (Reverte, 2009; Pavlopoulos, Magnis and Iatridis, 2017; Gupta, Krishnamurti and Tourani-Rad, 2018; Mohamed Adnan, Hay and van Staden, 2018).

Hence, drawing upon extant literature, we envisage that CGQ assumes a pivotal role in fortifying the relationship between IRQ disclosure and ICC in developed markets. De Villiers et al. (2017) introduced IRQ as an approach to reporting that encompasses both financial and non-financial information within a unified report. Subsequent extensive research has investigated the correlation between IRQ disclosures, CGQ, and ICC. Consistently, studies have revealed a positive association between IRQ disclosure and

---

## [Decision Letter · Decision Letter 1]

22 Apr 2025

Effect of integrated reporting quality disclosure on cost of equity capital in developed markets: Exploring the moderating role of corporate governance quality

PONE-D-24-26051R1

Dear Dr. Sarkar,

We’re pleased to inform you that your manuscript has been judged scientifically suitable for publication and will be formally accepted for publication once it meets all outstanding technical requirements.

Kind regards,

Yongliang Yang, Ph.D.

Academic Editor

PLOS ONE

Additional Editor Comments (optional):

Reviewers' comments:

Reviewer's Responses to Questions

**Comments to the Author**

1. If the authors have adequately addressed your comments raised in a previous round of review and you feel that this manuscript is now acceptable for publication, you may indicate that here to bypass the “Comments to the Author” section, enter your conflict of interest statement in the “Confidential to Editor” section, and submit your "Accept" recommendation.

Reviewer #1: All comments have been addressed

2. Is the manuscript technically sound, and do the data support the conclusions?

Reviewer #1: Yes

3. Has the statistical analysis been performed appropriately and rigorously? 

Reviewer #1: Yes

4. Have the authors made all data underlying the findings in their manuscript fully available?

Reviewer #1: Yes

5. Is the manuscript presented in an intelligible fashion and written in standard English?

Reviewer #1: Yes

6. Review Comments to the Author

Reviewer #1: The authors have managed to address all the recommendations made in the previous review, and in my opinion, the current version of this manuscript is fit for publication.

7. PLOS authors have the option to publish the peer review history of their article (what does this mean? ). If published, this will include your full peer review and any attached files.

**Do you want your identity to be public for this peer review?** For information about this choice, including consent withdrawal, please see our Privacy Policy .

Reviewer #1: **Yes: ** Cláudia Pereira

---

## [Editor Report · Acceptance letter]

PONE-D-24-26051R1

PLOS ONE

Dear Dr. Sarkar,

I'm pleased to inform you that your manuscript has been deemed suitable for publication in PLOS ONE. Congratulations! Your manuscript is now being handed over to our production team.

Kind regards,

on behalf of

Associate Professor Yongliang Yang

Academic Editor

PLOS ONE